# Viscoelasticity of diverse biological samples quantified by Acoustic Force Microrheology (AFMR)
Giulia Bergamaschi [1,2], Kees-Karel H. Taris [1,2], Andreas S. Biebricher[1], Xamanie M. R. Seymonson [1], Hannes Witt [1], Erwin J. G. Peterman [1] & Gijs J. L. Wuite [1] ✉

In the context of soft matter and cellular mechanics, microrheology - the use of micron-sized particles to probe the frequency-dependent viscoelastic response of materials – is widely used to shed light onto the mechanics and dynamics of molecular structures. Here we present the implementation of active microrheology in an Acoustic Force Spectroscopy setup (AFMR), which combines multiplexing with the possibility of probing a wide range of forces ( ~ pN to ~nN) and frequencies (0.01–100 Hz). To demonstrate the potential of this approach, we perform active microrheology on biological samples of increasing complexity and stiffness: collagen gels, red blood cells (RBCs), and human fibroblasts, spanning a viscoelastic modulus range of five orders of magnitude. We show that AFMR can successfully quantify viscoelastic properties by probing many beads with high single-particle precision and reproducibility. Finally, we demonstrate that AFMR to map local sample heterogeneities as well as detect cellular responses to drugs.

The cytoskeleton, a network of actin, microtubules and intermediate filaments, is the main determinant of cellular mechanics[1], contributing to the maintenance of cellular shape and resilience. In addition, mechanical properties of cells are strongly dependent on their surroundings including the extracellular matrix (ECM), a network of fibrous proteins with collagen as the main structural element. The ECM also contributes, together with external physical cues and substrate mechanics[2,3], to the regulation of cellular stiffness. Mechanical properties of cells, in turn, play a crucial role in directing cellular functions[2,4] and behavior, such as migration[5], differentiation[6], growth and morphology[7].

Biological soft matter exhibits rich physical behavior, such as the capacity of displaying both solid-like elasticity as well as liquid-like viscosity (generally termed viscoelasticity) and non-equilibrium phenomena[8,9]. Therefore, probing of viscoelastic properties over a wide range of frequencies (i.e. timescales) is a valuable tool to study the kinetics of biological processes of interest[10]. Furthermore, studies on viscoelasticity (generally referred to as rheology) can also be used to differentiate between healthy and diseased biological matter. For example, dynamic changes in viscoelasticity have been demonstrated to be a hallmark of aging[11], as well as diseases including cancer[12–14] and inflammation[15]. Similarly, disease-caused structural abnormalities in extracellular matrix proteins[16] can also be identified by rheological methods.

Traditionally, rheology is performed in bulk using rotational rheometers, which suffer from instrument inertia at high frequencies and typically require large sample volumes (order of milliliters[17,18]). Additionally, while rheometers are well suited for studying polymer networks, it is very difficult to obtain information on individual cells because they are either present in a monolayer[19,20] or embedded in an ECM-like gel, and are sheared collectively[21].

More recently, microrheology—or single-particle rheology—has been developed, which makes use of micron-sized probes to measure on substantially smaller length scales. Microrheology thus allows probing microscopic systems such as single cells and local network compartments, thereby offering direct insight into local heterogeneity and thus bypasses the drawbacks of traditional bulk rheology[17,22]. A direct comparison between the rheological properties of entangled networks and living cells has enabled the application of single-filament and network-dynamics theories to microrheological data on living cells obtained via Atomic Force Microscopy (AFM) or optical tweezers (OT)[23–25], paving the way to a more complete understanding of complex system mechanics.

Single-particle rheology can be subdivided into active and passive microrheology. While passive microrheology is based on the tracking of particles driven by Brownian motion, active microrheology involves the application of an external oscillating force to extract viscous and elastic

[1]Department of Physics and Astronomy and LaserLaB Amsterdam, Vrije Universiteit Amsterdam, Amsterdam, The Netherlands. [2]These authors contributed equally: Giulia Bergamaschi, Kees-Karel H. Taris. ✉e-mail: g.j.l.wuite@vu.nl

https://doi.org/10.1038/s42003-024-06367-3                                                                                                   **Article**

moduli of the samples. Active microrheology can also mimic stresses often encountered in vivo[26,27], while the presence of an external force reduces the noise level of the traces[8].

In recent years, active microrheology has been conducted using a variety of different techniques including magnetic tweezers (MT[28,29]), OT[30,31], and AFM[14,24,32]. While each approach has its own advantages, typical limitations include lack of multiplexing capability (AFM and OT) or a limited range of attainable forces (MT < 0.1nN; OT <1nN[33,34]).

AFS is an emerging technique that makes use of a piezoelectric element to generate high-frequency (MHz) standing waves in a microfluidic chamber. The resulting acoustic field allows the application of a wide range of forces (few pNs to tens of nNs) to compressible spherical particles such as polystyrene or silica beads[35,36]. While this technique is fairly new, the general method of using acoustic fields to apply physical stresses is long established[37,38] and its physical foundations are very well understood. AFS combines the multiplexing capabilities of MT with the stress range of AFM and OT, and measurements can be readily conducted in a microfluidic chamber, which allows for quick buffer exchange and/or application of fluid stresses[39]. So far, AFS has mainly been employed to assess the mechanical properties of living cells[39–41], but most of these approaches did not yield complex moduli or frequency-dependent mechanical properties. Only very recently, AFS has been used to perform multi-frequency microrheology on a monolayer of endothelial cells[42], establishing the capability of this approach to measure changes in elastic and viscous properties in a multiplexed fashion. This implementation was only demonstrated, however, within a very narrow frequency range (0.5–1.5 Hz) and applied to a single biological system.

Here, we develop acoustic force microrheology (AFMR) of solid materials by implementing application of oscillating forces ( ~ pN to ~nN) over a frequency range of five orders of magnitude (0.01–100 Hz). Furthermore, we implemented a synchronization scheme between force and distance readout, which proved crucial for proper phase determination at higher frequencies. We tested our AFS approach on biological systems of different nature and size, ranging from a collagen network to individual RBCs and human gingival fibroblasts. Measurements on well-characterized collagen networks allowed us to test the reproducibility and robustness of our approach, as well as the capability to detect local heterogeneity. Furthermore, using a simple single-cell system such as RBCs, we investigated the influence of varying pre-stresses and modulation amplitudes on their complex modulus. Finally, we probed the viscoelasticity of human fibroblasts and demonstrated that the method is able to discriminate between healthy and cytochalasin-D-treated fibroblasts.

## Results
### AFMR provides stable and reproducible viscoelasticity characterization

Our AFS set-up (based on[36]) consists of a microfluidic chamber inside a glass chip with a glued-on (transparent) piezoelectric actuator which is controlled by a function generator. If the actuator is oscillated at the resonance frequency of the chamber ($f_{res}$ = 14.2–14.4 MHz), the ensuing acoustic standing wave will push the micron-sized spherical probe particles in the field-of-view towards the node with a force that depends on the applied voltage amplitude square (see Methods). By modulating the piezo amplitude of the resonance carrier wave with a sine wave of lower frequency, we are thereby able to generate an oscillation of the force acting on the beads. Calibration of applied forces is achieved using the shooting-up method (Supplementary Fig. 1a), in which the force is deduced using the Stokes-drag relation of a particle of known size in a Newtonian fluid of known viscosity, in our case the standard buffer (see Methods). The acoustic field always displays some local heterogeneity in strength, which we did not correct for in the current measurements. It has previously been shown that for samples having similar compressibility as water (such as cells or diluted gels), such correction does not significantly impact average values[43]. In addition, potential heterogeneity in acoustic field is not expected to affect parameters

such as power-law exponents (see Methods). Even though the force mainly acts in the vertical (z-)direction, we still observe horizontal (XY-) deflections in some cases; these, however, have no influence on the measurement results (Supplementary Fig. 1b).

The glass chip is mounted on an inverted microscope equipped with an LED for brightfield illumination (bLED) and the bead displacement is recorded on a CMOS camera via an objective (Fig. 1a). Since the precise determination of the phase shift between the applied force and the resulting probe movement constitutes a key element in active microrheology experiments, it is of the utmost importance to ensure synchronized force application and distance detection, as has been used in OT setups[44]. However, we found that this required a slight modification of the original set-up (see Methods). Thus, we introduced a second, low-latency LED (sLED, response time «1 ms), the intensity of which oscillates in phase-lock with the piezo-drive signal and thus provides a time stamp light-intensity signal (Fig. 1b). Since the CMOS camera then detects both the input (=sLED intensity) as well as the response (=bead displacement) oscillation, this approach circumvents any instrument-related delay. Elastic (G') and viscous (G") moduli were calculated from these raw traces by determining the frequency transfer function between the intensity (force) input and the displacement output to obtain the phase shift between the two signals, whereas the amplitudes were extracted from the Fourier transforms of the signals (see Methods). Traces with non-periodic, irreversible length changes (such as ruptures) were excluded from the analysis. Furthermore, while active microrheology also allows probing non-linear responses such as strain-stiffening, our data here mainly covers the linear regime (see "Results" section).

Similar to MTs, AFS can only apply unidirectional forces, which stipulates that for all measurements a minimum pre-stress equaling the oscillation amplitude $A_F$ has to be applied. Thus, our data considers two different force oscillation parameters: (1) the pre-stress value $F_p$ as the offset around which the force oscillates ($F_p \geq A_F$), and (2) the so-called modulation depth, defined as $MD = A_F/F_p * 100\%$. Another side effect of unidirectional force application is that AFMR characterization of liquids is not possible, which is why our study is limited to viscoelastic solids. This has the advantage that the size of the probe particle is of minor importance and only relevant when computing the complex modulus' prefactor ([22]; Methods). When probing gels, the particles have to be larger than the mesh size to prevent slipping (see "Methods").

In order to test the general capabilities of our experimental set-up, we first conducted AFMR experiments on polystyrene bead embedded within a collagen gel (Fig. 1c). Since the acoustic field acts on all microspheres in the chamber simultaneously (Fig. 1a) and we can track beads over a large range of focal distances ( > 10 µm in z-direction, see Methods), the multiplexing capabilities of our approach are mainly limited by the required tracking resolution (which affects the objective choice 10x to 40x, see "Methods") and the acquisition rate of the camera (30–300 Hz). This means that we can successfully measure between 5 and 100 beads in parallel depending on the sample and maximum oscillation frequency (see Methods). While increasing bead density will also result in a higher number of probes that can be measured simultaneously, we abstained from using too high densities in order to prevent beads in close proximity interfering with each other (Supplementary Fig. 1c shows this was not the case in the results presented here).

Typical raw traces of distance and force plotted against time (Fig. 2a) provide direct testimony to the softness of the assembled collagen gel, deducible from the fact that an oscillation amplitude of <10 pN results in a bead displacement of >1 µm. The increasing phase delay between distance and force oscillation is not only visible directly in the raw traces but can be even better visualized as the hysteresis in the displacement vs. force plot (resulting in a so called first-order Lissajous figure, Fig. 2b). These also show that the enclosed areas mostly follow an elliptical shape; this, together with force clamp tests (Supplementary Fig. 1d), confirmed that we probe in the linear-response regime. The phase delay directly reports on the relative strength of loss vs. storage modulus and signifies a more viscous response on

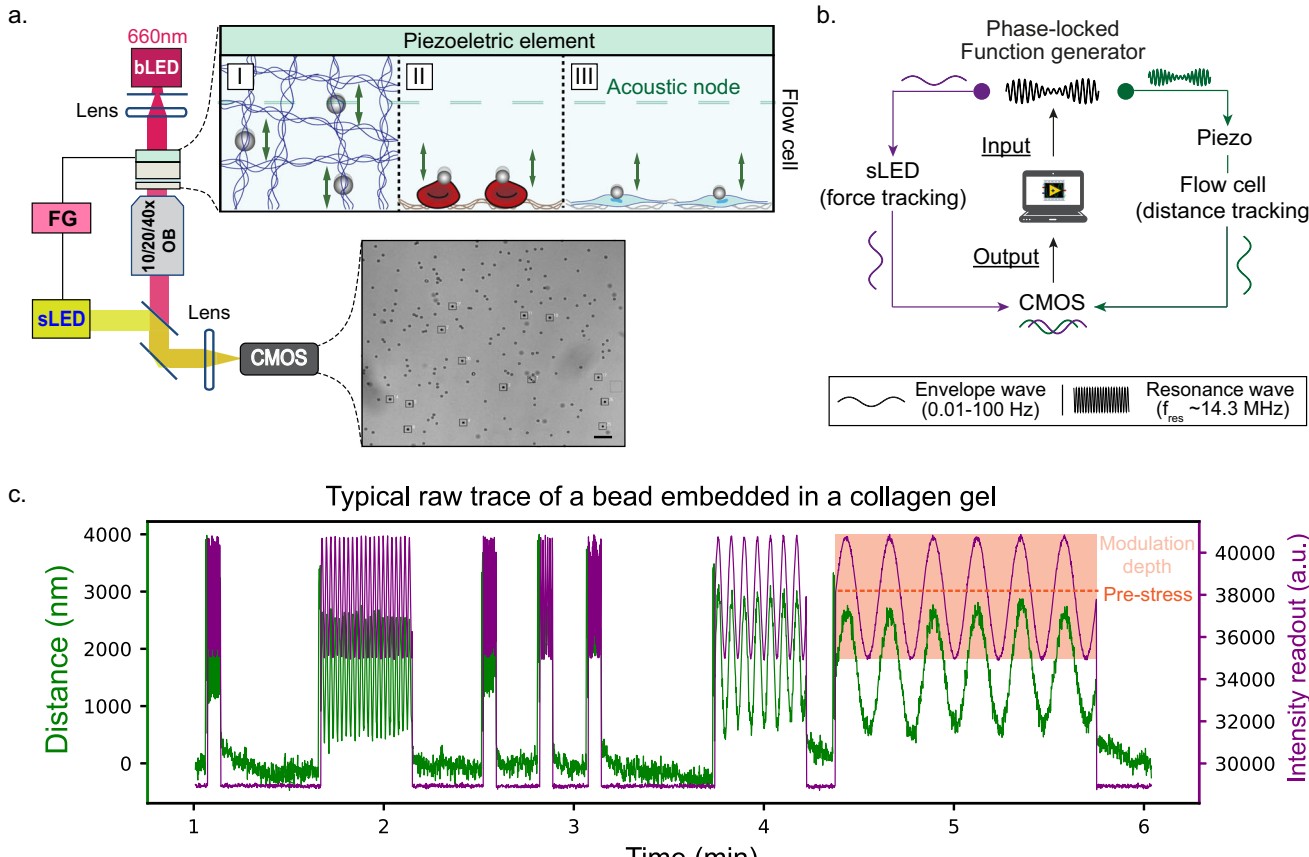

**Fig. 1 | Experimental setup and force application method of AFMR. a** Scheme of the AFMR-setup: A piezoelectric element mounted on a glass chip allows generation of an acoustic field on beads located in the microfluidics chamber. The scheme in the top inset depicts the three probed sample types: (I) collagen gel or (II) surface-adhered RBCs and (III) HgF. Imaging of the sample is achieved using a constant-intensity brightfield LED (bLED) that images the sample via an objective onto a CMOS camera. The typical example image (inset bottom) shows a typical field of view (FOV) of beads embedded in a collagen gel, with the individual measured beads selected for tracking enclosed in a square. Scale bar is 20 μm. A second, synchronisation LED (sLED), with an intensity that is oscillated in sync with the force application ensures synchronized detection of force and particle displacement. **b** Hardware block diagram that depicts how AFMR enables signal synchronization. The function generator sends out a sine oscillation signal of desired frequency, phase-locked to two different components: (I) the synchronisation LED (sLED,

green) and (II) the piezo-element attached on top of the AFS flow cell (purple, note that for (II), this signal is impressed on the resonance frequency of the chamber by amplitude-modulation, since the latter generates the acoustic force). Since this enables the detection of both the force (sLED) and the distance (bLED) signal via the same hardware (CMOS), intrinsic synchronization of both signals is guaranteed. **c** Typical raw trace of Intensity and height over time of a single bead embedded in a collagen I gel (~ 0.2 mg/ml); in this experiment, frequencies of 1, 0.3, 3, 10, 30, 0.1, 0.03 Hz were applied around a pre-stress of ~13pN with a modulation depth (i.e. the oscillation amplitude normalized by the offset force) of 50% ( ~ 7pN). Note that the raw data shows z-distance and sLED intensity as a function of time, since the modulation of the latter is used as the force readout. The sLED intensity can be readily converted to the corresponding force after voltage calibration (see "Methods").

shorter timescales, a feature that is observed for most biological visco-elastic materials[45].

Given that traces from single beads yielded convincing results, we next set out to validate the reproducibility of our results and estimate the accuracy of single bead measurements (Fig. 2c, d). To this end, we repeatedly measured the same beads over time; collagen networks are well suited for this since they show little dynamics, preventing their specific viscoelastic properties from changing over time. Figure 2c shows calculated G' & G" values for an exemplary bead measured at five different time points. From the resulting moduli variance, we deduce a relative standard deviation (s.d.) below 20% (e.g. RE(G"$_{1Hz}$) ~ 15%, G"$_{1Hz}$ = 0.050 ± 0.007 Pa, Supplementary Table 1), then repeated this procedure for all the beads in a given FOV and found similar relative s.d. values for both G' and G" (10% < RE < 20%) for most of the frequencies examined, with slightly larger variances being observed only at the lowest/highest frequencies of G' & G", respectively (Fig. 2d). Note that these s.d. values are determined for a soft collagen gel, such that small absolute errors lead to relatively large relative errors. Nevertheless, these serve as an upper bound for the instrument-

related error and will become relevant for differentiating between instrument-related noise and natural variance of the studied system (see below and Methods).

## AFMR of collagen I gels shows local heterogeneity and decreasing viscosity at higher collagen concentrations

We next used AFMR to investigate the effect of collagen gel concentration (Fig. 3a, b) and observed a significant stiffness increase with concentration (extracted low-frequency moduli of G'$_{0.03Hz}$ = 1.67 ± 0.10 (s.e.m.) Pa and G'$_{0.03Hz}$ = 0.13 ± 0.02 (s.e.m.) Pa for 0.6 mg/ml and 0.2 mg/ml collagen concentration, respectively), which is expected[46]. Although collagen stiffness is known to be very sensitive to the self-assembly environment[47,48], our stiffness value of G'$_{1Hz}$(0.6 mg/ml) = 2.7±0.1 Pa is similar to G'$_{1Hz}$(0.5 mg/ml) = 3.9 Pa reported under comparable conditions[49]. In addition, we confmed that the frequency dependence of the viscous modulus decreases as the concentration increases[46]. Thus, when we fitted the average G"-values with a power law distribution (G" ~ f$^\alpha$) at frequencies above 0.3 Hz, we found exponents of α$_{low}$ = 0.60 ± 0.02 and α$_{high}$ = 0.25 ± 0.03

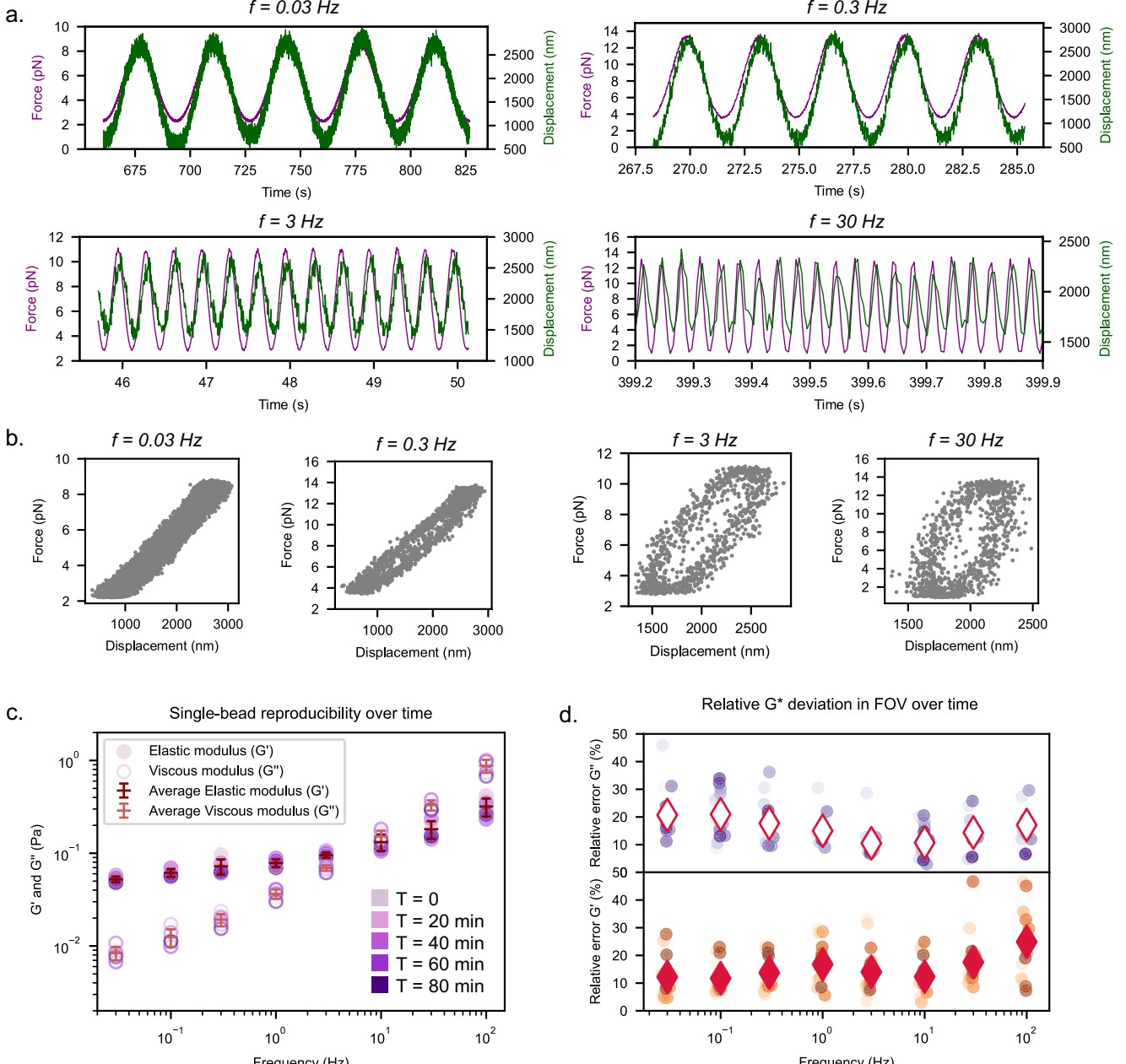

**Fig. 2 | Characterisation of AFMR method and reproducibility using polystyrene beads embedded in a collagen gel. a** Typical raw traces of 6.39 μm polystyrene beads oscillated in a collagen gel (concentration ~0.2 mg/ml) for four different frequencies, recorded using a 10x objective at a frame rate of 80 Hz. Note that the force amplitude is adjusted for each frequency in order to ensure sufficient bead displacement while staying in the linear regime (see "Methods" and Supplementary Fig. 1d). Note that the trace fluctuations are less pronounced in other samples (such as RBCs, Supplementary Fig. 2a), thus we attribute them to the softness of the gel rather than to instrumental noise. These fluctuations also do not impact the FFT analysis (Supplementary Fig. 1e). **b** Typical force-displacement (Lissajous) plots for the raw traces shown in (**a**). The increasing phase shift between force and displacement is readily visible from the decreasing overlapping of signals, and larger enclosed areas at higher frequencies in the Lissajou plots. While the plots can slightly deviate from a perfect ellipsoidal shape, our control data indicates that we still probe the linear regime of the sample (Supplementary Fig. 1d). **c** Single-bead ($n = 181$; $N = 2$) reproducibility over time at timepoints t = 0,20,40,60 and 80 min. An example of a single collagen-embedded bead measured repeatedly over time applying frequencies from 0.03 to 100 Hz. Average G-values and the corresponding s.d. of repeated measurements computed for each frequency are shown as red lines. **d** Relative deviations of G' and G" considering repeated measurements on all beads in a FOV. Average relative errors (RE, diamond markers) on both G' and G" are mostly lower than 20%, and higher errors (RE-G"$_{0.03Hz}$ = 21%, RE-G'$_{100Hz}$ = 24%) are found for the extremes of the frequency spectrum: 0.03 Hz and 100 Hz.

for the low and high gel concentrations, respectively. This agrees with previous reports of only little frequency dependence for collagen I gels of even higher concentrations (>3 mg/ml;[46,50,51]).

Since microrheology uniquely allows the study of viscoelastic heterogeneities within an assembled network[49,52], we investigated the variance of viscoelastic parameters for the two different collagen concentrations. Here, we focused again on the frequency dependence of the viscous modulus, also

since this value is not influenced by the local heterogeneity of the acoustic field[42,43] (see "Methods"). Nevertheless, in order to better appreciate the sample heterogeneity, this time we considered the frequency dependence of G" for each bead individually. This analysis demonstrated not only that the frequency dependence for a single bead also follows a power law, but that for the low-concentration gel (Fig. 3c) the power law exponent can vary considerably from bead to bead, in agreement with a previous report[49]. While we

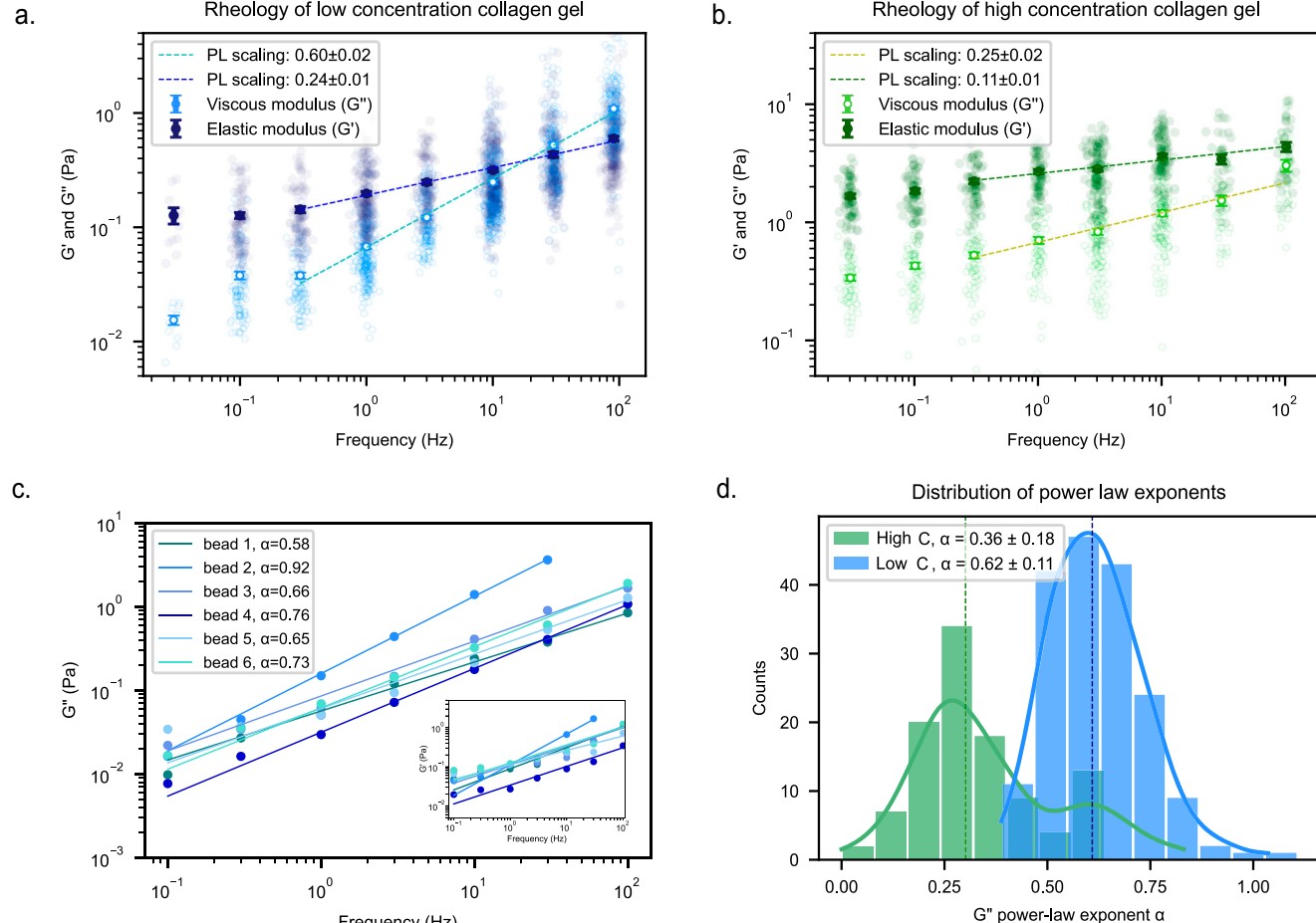

**Fig. 3 | AFMR measurements of collagen gels at two different concentration.** G'
and G" moduli computed for (**a**) low concentration (0.2 mg/ml) and (**b**) higher
concentration (0.6 mg/ml) collagen gels ($N_{low}$ =181 (2), $N_{high}$ = 139 (2)). Power-law
fits of mean data yield G"-exponents of $\alpha_{low} = 0.60 \pm 0.02$ (s.d.) and
$\alpha_{high} = 0.25 \pm 0.03$, respectively. Note that despite the G"-data spread, the large
number of data points results in very low s.e.m. values of the mean moduli, which in
turn accounts for the low variance of fitted exponents. **c** The local heterogeneity for
the lower concentration collagen gel can be visualized by comparing the slope
variance of the viscous modulus G" between different beads; a power-law scaling is

applied yielding the power-law exponents ($R^2 > 0.98$). Inset shows that similar
heterogeneity in the frequency dependence can be seen for G' as well. **d** Histogram of
the power-law exponents for each individual bead in high (green) or low (blue)
concentration collagen gels; lines highlight the probability distributions (KDE),
dashed vertical lines show the average value. Via a two-tailed Mann–Whitney sig-
nificance test, $p < 0.0001$ (****) was obtained. Note that here the results of single-
bead power-law fits are used, in contrast to (**a**, **b**) where power-law scaling were
performed on the mean and s.e.m. of multiple beads ($N_{low}$ =181 (2), $N_{high}$ = 139 (2)).

observed similar characterics for the storage modulus (inset Fig. 3c), we
focused here on the G"-data analysis, since the exponent is in this case much
larger (Fig. 3a), thus should provide more robust data.

In order to quantify this observed heterogeneity, we first estimated the
single-bead variance in the frequency dependence, analogous to the
approach used in Fig. 2d. Therefore, we calculated the s.d. of the power law
exponent obtained from repeated measurements of the same bead (Fig. 2c)
and used this to determine an error (s.d.) for a single exponent of about ~6%,
when averaged over separate single bead measurements (see Methods and
Supplementary Fig. 1f). Similar to what we stated above, this value should
provide an upper bound for the variance that can be accounted for by
instrument noise. We then fitted histograms of the power-law exponents of
all beads for both collagen gels (Fig. 3d) by a Gaussian distribution, and
obtained averages and corresponding s.d. values of $\alpha_{low} = 0.62 \pm 0.11$ and
$\alpha_{high} = 0.36 \pm 0.18$. Both exponents are as expected very similar to those
deduced from fitting the frequency-averaged data above (Fig. 3a, b), but the
much larger variance demonstrates that the second approach provides a
much better measure of the sample heterogeneity. This is confirmed by the
observation that the corresponding relative s.d. values of RE($\alpha_{high}$) $\approx 50\%$
and RE($\alpha_{low}$) $\approx 18\%$ are much higher than the upper bound of the instru-
ment error (6%) which we deduced above, therefore highlighting the

potential of this approach to unravel local viscoelastic differences within a
network.

## RBCs show increasing viscosity with frequency, while high pre-stresses and modulation amplitudes result in non-linear behavior

For the application of AFMR to single cells, we first studied human RBCs,
which are structurally relatively simple cells: they do not have a nucleus and
can be modeled as a filamentous protein network confined within a lipid
bilayer[53–56]. Mechanical characterization of RBCs is biologically relevant,
since they are in vivo constantly subjected to stresses caused by the turbulent
flow through thin blood vessels. Their deformability is also of clinical
importance, since it is impacted in diseases, such as anemia[57], diabetes[58],
malaria[59,60] and viral infection from Covid-19[61]. For our experiments, we use
relatively large (6.59 μm) beads compared to previous OT or MT studies
(<4 μm;[34,62,63]), which allow application of stresses >5 Pa, well within the
physiologically relevant range of 1–20 Pa[64,65]. Moreover, in comparison to
earlier microrheology studies, in which only local stresses («1 μm²) were
exerted on the RBCs, we are able to apply stresses over much larger areas
(~ 8 μm²). We thereby expect to be closer to in vivo conditions, where red
blood cells deform globally rather than locally when being squeezed through
narrow capillaries.

While RBCs are considered to be softer than many other cells, their stiffness is still several order of magnitude larger than that of collagen gels, therefore an oscillation amplitude of 100 pN will typically result in displacements <1 μm (Supplementary Fig. 2a, b). The frequency dependence of the moduli is also much less pronounced compared to the collagen system, and for frequencies up to 1 Hz, our data show a constant storage modulus $G'_{0.01Hz}$ of 72±4 Pa (Fig. 4a), in the lower end of the wide range of previously reported values of 0.06 to 3.3 kPa[64,66], and 5 kPa[67].

This can be due to different surface-attachment protocols used in some of the AFM studies, as well as different probe contact areas and pulling geometries[68]. Before, it has been reported that higher concentrations of poly-L-lysine, which results in stronger surface attachment, result in higher stiffness measured with AFS[40].

We found that the viscous modulus G″ -in contrast to G′- shows a pronounced frequency-dependence at frequencies above 1 Hz that can be described by a power law with an exponent α = 0.47 ± 0.05 (Fig. 4a). Our value is on the lower end of previously reported exponent values between 0.5 and 0.64[34,69]. Interestingly, at f < 1 Hz, G″ seems to show different frequency (time) dependencies, which hint at >2 distinct relaxational timescales for this system[70]. This is in line with recent observations of two sub-1Hz relaxational processes in RBCs (~0.01, 0.2 Hz;[71]).

Whereas the G″-exponent for the RBC data above yields a quite robust average value, we learned from the corresponding collagen data (Fig. 3a, b) that the fit error yields a poor insight into the sample heterogeneity, which is also clearly visible from the data spread of the RBC data (Fig. 4a). We therefore used also here a very similar approach as used for the collagen data, and instead fitted individual bead data by a power-law relation. The corresponding distribution of alpha values (Supplementary Fig. 2c) for the sample can be nicely fitted by a Gaussian distribution, yielding a mean value of 0.63 ± 0.14 (s.d.). While the average value is expectedly still in the range reported for RBCs, the s.d. value is now much larger and corresponds to a RE value of ~21%. This is notably larger than the ~6% we deduced above to constitute an upper bound of the instrument-related uncertainty (see Methods). Therefore, this result would indicate that quite similar to the collagen gel, we observe a bead-to-bead variance intrinsic to the sample, with the main difference that this time we do not resolve a spatial, but rather a cell-to-cell heterogeneity. We additionally conducted AFMR experiments where we measured the same RBC batch repeatedly (4x times over the time span of 1 h, Supplementary Data 2–3). While the resulting distribution (Fig. 4b) does show some data variance, the resulting overlap of the complex moduli over time would indicate that our RBC sample does not show significant fatigue-induced stiffening[72] within the duration of the experiments.

Since our set-up allows tuning of both modulation depth and pre-stress over a large range, we next tested to what degree these parameters affect the viscoelastic properties of RBCs. To this end, we first measured the complex moduli at a constant pre-stress value of ~110 pN while varying the modulation depths between 25 and 100% (corresponding to force amplitudes of 28–110pN; Fig. 4c). Nevertheless, as expected for a linear response, the complex modulus |G*| was not significantly affected by the change of modulation depth at this rather low prestress. Additionally, we tested the effect of increased pre-stresses while keeping the modulation depth constant at 50% (Fig. 4d). For low to moderate pre-stresses (100–300 pN; used range in the experiments presented here), we did not observe a large increase ( > 20%) in the measured moduli, as expected within the linear regime. Comparing these to the largest pre-stress value, we did find—despite the cell-to-cell heterogeneity - an increase of G* with pre-stress for individual cells (Supplementary Fig. 2f). This indicates that cells start to become stiffer at higher tension and therefore constitutes stress-stiffening behavior, which is a common property of biological materials[19,73], including RBCs[34,69,70,74]. Overall, these results demonstrate that AFMR is suitable for probing both the linear and non-linear viscoelastic response. We found that the global RBC deformation demonstrates a linear response over a wide range of distance amplitudes (MD < 75%) at pre-stresses <300 pN. Overall, the RBC

experiments highlight the capability of AFMR to perform accurate measurements on a simple single-cell system as well.

## AFMR of Human gingival fibroblasts shows elastic response, that decreases upon cytochalasinD treatment

After characterizing RBCs as simple model cells, we performed AFMR on human gingival fibroblasts (HgF), as complex, adherent yet motile cells, the viscoelastic properties of which have not yet been studied. Measurements were performed on cells adhering to a collagen-coated flow cell surface, using 6.59 μm diameter silica beads as a probe. Raw traces show that force amplitudes of ~500 pN are required to cause substantial bead displacement (i.e. >30 nm, Supplementary Fig. 3a, b), in agreement with previously reported AFM[75] and micropipette aspiration studies[76] on other fibroblast cell lines, indicating that HgF are substantially stiffer than RBCs.

Apart from the increased stiffness (Fig. 4e), our AFMR results on fibroblasts show predominantly elastic behavior at lower frequencies (<10 Hz), a trait shared with RBCs. Furthermore, storage and loss moduli are only slightly frequency dependent and the plateau modulus $G'_{0.1Hz}$ at lower frequencies is 1.92 ± 0.21 kPa, in line with previous AFM studies on fibroblast cells[24,32]. Our results also reproduce earlier reports that the complex modulus can be fitted with a structural damping model, from which a scaling exponent of α = 0.10 ± 0.01 was retrieved, thus within the range of literature values on fibroblasts (0.05 < α < 0.35[10,24,77,78]). Finally, we tested whether AFMR is capable to resolve changes in cell mechanics caused by treatment with the actin-depolymerising drug cytochalasinD. Following treatment with cytochalasinD, cells showed clumped cell-bodies and thinner protrusions (Supplementary Fig. 3c;[79,80]) and both loss and storage moduli decreased (Fig. 4f). Thus, after treatment, the low-frequency plateau modulus was measured at $G'_0$ = 0.84±0.08 kPa, a 2.3-fold decrease compared to untreated HgF. Finally, we observe that the power-law exponent increased slightly (α = 0.13 ± 0.01) upon treatment with cytochalasin, as has been reported before[10,64,80,81]. Altogether, we demonstrate here that AFMR is also capable of accurately probe stiff motile cells, and detect changes in their viscoelastic properties.

## Discussion

In this study, we present AFMR, a method for performing active microrheology on biological and synthetic samples using AFS. The set-up we developed ensures intrinsic synchronisation of force and distance detection, and permits a large force ( ~ 1–10⁴ pN) and frequency (0.01–100 Hz) range.

AFMR combines several advantages of other microrheology techniques in one: it allows multiplexing/parallel measurements comparable to MT, while it can apply high forces (up to tens of nN) similar to AFM. We demonstrate this unique combination of abilities by employing AFMR to characterize a wide range of samples, spanning a viscoelastic modulus range of five orders of magnitude. We show that it is equally suited to study the local heterogeneity of a collagen gels—previously only studied by considering a few locations in the sample—assembled in vitro, to investigate the viscoelastic change of RBCs, and to finally detect drug-induced mechanical impairment of stiff adherent HgF cells, which had not been mechanically characterised before. This wide application range is also afforded by its capability to accurately determine viscoelastic properties measured from a single bead. While microrheology data on single beads have been reported before, our results are noteworthy since they can assign an upper bound of the instrument error in the data (see Methods), which is vital to identify a significant deviation of viscoelastic parameters between different datasets.

For these reasons, we propose that AFMR is a particularly attractive method for studying local network heterogeneities, cell-to-cell variance and cell dynamics. Our results on local viscoelastic heterogeneities of collagen gels and cell-to-cell variance of RBCs, providing insight into a subject that has seen only few experimental studies. Specifically, while local heterogeneity in networks has been studied before, this has only been accomplished by passive microrheology of diffusive particles embedded in non-crosslinked networks; there, heterogeneities were identified as deviations from a Gaussian displacement distribution[82–84]. Nevertheless, studies of

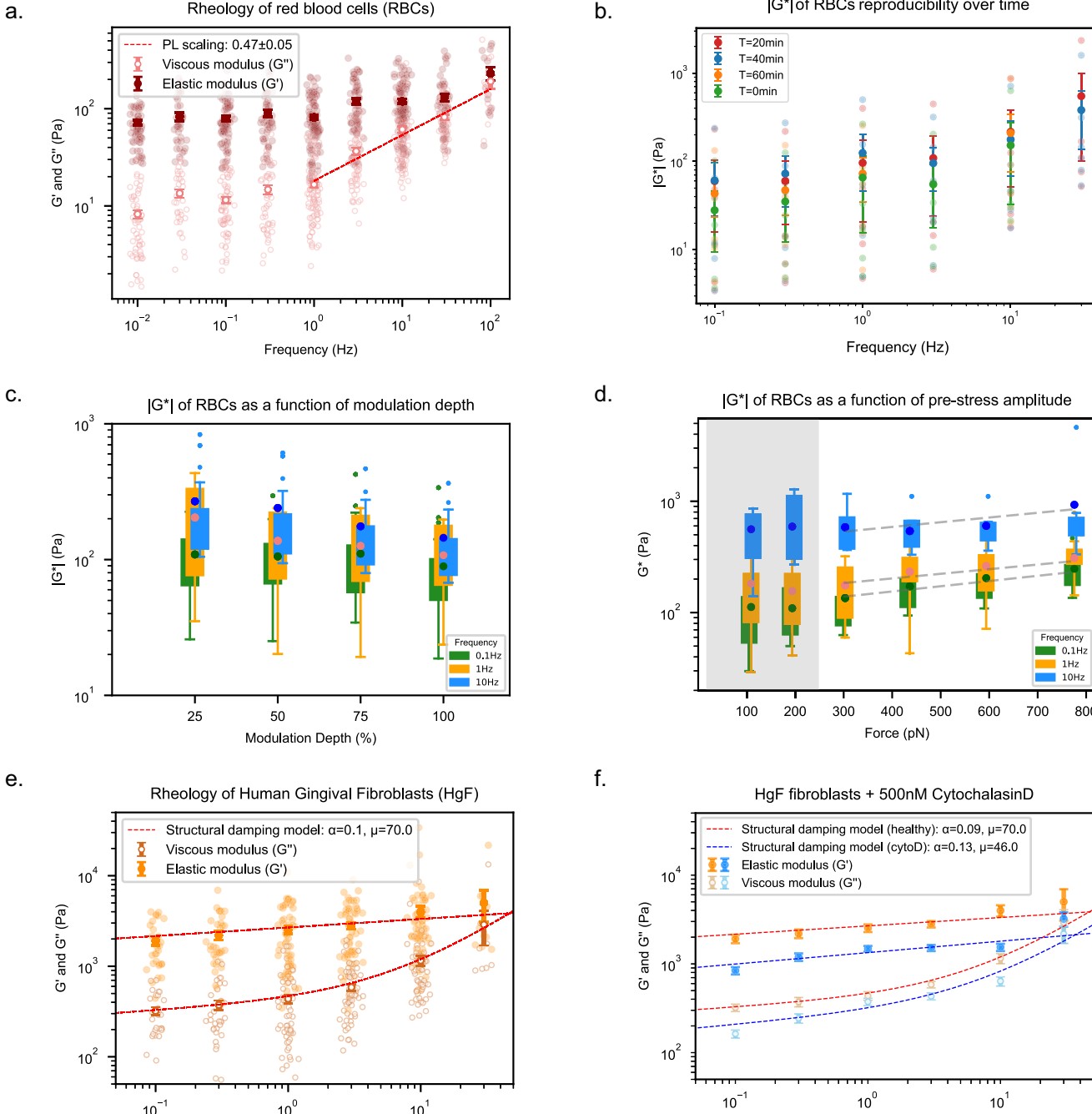

**Fig. 4 | AFMR of RBCs and HgF under different force modulation settings and experimental conditions. a** Single-cell rheology: G' and G" moduli determined for RBCs ($n = 143$; $N = 11$). Error bars are SEM; for most frequencies they are smaller than the datapoints. A power-law scaling performed on the G" data yields an exponent $\alpha = 0.47 \pm 0.05$. Error is determined from the square root of the fit's covariance matrix. **b** Average complex modulus with s.d. of a FOV of RBCs measured over one hour ($n = 5$–$5$; $N = 1$). **c** Effect of modulation depths (amplitudes) on the complex modulus G* of single red blood cells; selected frequencies of 0.1 Hz ($n = 24$; $N = 2$), 1 Hz ($n = 19$; $N = 3$) and 10 Hz ($n = 17$; $N = 2$) are shown for MD = 25%, 50%, 75%, 100%. Error bars are S.D. **d** Effect of higher pre-stresses on the complex modulus G* of single red blood cells; selected frequencies of 0.1 Hz ($n = 14$; $N = 4$), 1 Hz ($n = 19$; $N = 6$) and 10 Hz ($n = 10$; $N = 4$) are shown for

increasing pre-stresses. Error bars are S.D. The shaded area (gray) represents the force range used for RBCs experiments. Within this force regime, G* is constant. Power law fits ($\alpha_{0.1Hz} = 3.5 \pm 0.2$, $\alpha_{1Hz} = 7.1 \pm 0.3$, $\alpha_{10Hz} = 17.7 \pm 0.3$) highlight the stiffening region (Supplementary Fig. 2f). **e** Rheology of cultured HgF primary cells ($n = 71$; $N = 3$); a structural damping model fits well the G'-G" data ($R^2 = 0.98$). **f** Rheology of cultured HgF primary cells treated with 500 nM Cyto-chalasinD ($n = 32$; $N = 2$); a twofold decrease in both elastic and viscous moduli can be seen (95% CI$_{healthy}$ = 1701–4884 Pa, 95% CI$_{CytoD}$ = 957–1781 Pa), while the power-law exponent from the s.d. model increases slightly ($\alpha_{healthy} = 0.09 \pm 0.01$ vs. $\alpha_{cytoD} = 0.13 \pm 0.02$). Statistical significance was tested on G* data (HgF and HgF+cytoD) via a two-tailed paired $t$ test, $p < 0.02$ (*). Details of the test are in Statistics.

sample heterogeneity in viscoelastic solids using active microrheology is quite less common. Similarly, examining the full distribution of single-cell stiffnesses has previously been used to deduce variability in cell populations[85]. Here, we extend this approach to solid-gel systems, and we are also able to separate particle-to-particle variance from measurement noise. Since there is increasing evidence that mechanical heterogeneities both in gels[86] and cells[87,88] can give rise to intriguing and biorelevant properties, we are convinced that the future study of viscoelastic heterogeneities should likewise be of considerable interest to the biophysical community.

Finally, the possibility to tune both pre-stress and amplitude of modulation makes AFMR also highly suitable for the investigation of non-linear mechanical properties of materials: the standard protocol for such studies involves the application of small-amplitude oscillations on top of a large pre-stress[89]. Up to now, such experiments have mostly been performed on polymer networks using bulk rheology. With AFMR, a macroscopically large pre-stress can be applied, with the advantage that individual bead responses can be recorded. Moreover, it would be possible to measure time correlation functions when no force is applied, similar to OT and MT, as the distances are well within the AFS resolution[35,56]. All this combined makes AFMR a valuable asset when investigating a large variety of different (heterogeneous) samples. Thus, we expect AFMR to constitute an appealing methodology/technique for both soft-matter and cell-biology researchers alike.

## Methods

### Experimental AFS setup and signal processing

Our instrument (Fig. 1a) is based on a set-up described in an earlier publication[36], with a few modifications. In brief, we use a custom-made glass chip containing a single chamber (LUMICKS BV) mounted on a standard inverted microscope (Eclipse Ti, Nikon) and equipped with an air objective (Plan Fluor of either 10x/0.3, 20x/0.5 or 40x/0.75 magnification/NA; Nikon) and a piezoelectric xyz-translation stage (P-517.2CL, Physik Instrumente). A translucent piezoelectric actuator (LUMICKS BV) covering the whole chamber is glued on top and connected to a function generator (1032X, Siglent Techologies) which in turn is controlled by a custom-written LabVIEW program (https://doi.org/10.5281/zenodo.11115840[35]). Brightfield visualization of the beads is achieved by a constant-illumination LED (called here bLED; M455L3-C5, Thorlabs) with a constant intensity output, and transmitted light is imaged with a CMOS camera (DCC3240M, Thorlabs). Before each experiment, a height look-up table is created via the translation stage which in combination with a custom-written LabVIEW program ensures bead tracking in xyz as previously described[35,36]. This ensures accurate tracking of the beads (≤50 nm resolution, see below) over a height range of at least 10 μm.

This original AFS setup[36] was modified in order to allow for an intrinsic synchronization between force and distance signals. To this end, a synchronization low-latency («1 ms) LED (sLED; Thorlabs, MWWHF2) was used additionally for bright-field illumination, the intensity of which was oscillated (frequency range 0.01–100 Hz) phase-locked with the modulation wave feeding the piezo actuator. Therefore, the function generator sends the same modulated wave to two separate channels: piezo and sLED; however, in the first case, the oscillation signal is impressed on the resonance frequency of the glass chamber (14.2–14.4 MHz) which generates the acoustic field (Fig. 1b). The resulting brightness modulation is homogeneous over the whole camera FOV and can thus be followed by analyzing the intensity of a background ROI. Since the absolute background intensity is dependent on location and exposure time, we choose for every new FOV a new ROI (used for the sLED brightness readout) inside the same measurement FOV and recalculate the corresponding brightness-to-force conversion factor. The location or size of this ROI region does not influence the accuracy of the phase detection (Supplementary Fig. 1g). Both the bead tracking, executed as previously detailed using a quadrant-interpolation algorithm for xy-position and a lookup table for z-position[36], and the background intensity are then extracted from the same camera image using a custom written

LabVIEW program. Since in this case, the camera detects both the distance change (using the bead tracking via the bLED) and the force signal (synchronous with the brightness oscillation sLED), both of these signals are recorded intrinsically synchronized and without any artificial delays due to differences in the signal processing. The modulation frequencies used in the experiments ranged from 0.01 Hz to 100 Hz, while the duration of the oscillations was set such that enough cycles were recorded for a robust analysis. For f > 3 Hz, a duration of 10 s was used; this resulted in 30 cycles for e.g. a 3 Hz trace, and 300 cycles for a 30 Hz trace. For 0.1 Hz < f < 1 Hz, the duration was set to 50–100 s, resulting in e.g. 10 cycles for a 0.1 Hz trace, up to 50 cycles for 1 Hz oscillations. Low frequencies oscillations, such as 0.01 Hz and 0.03 Hz, had the lowest number of repeats (3–5 cycles) in order to limit the time required for data collection.

While function generator and piezo would be capable to support force oscillation frequencies up to >1 kHz, in practice we are limited by the acquisition rate of the used camera. Thus, in full FOV mode, the camera achieves a maximum frame rate of ~60 Hz, which limits oscillations to 10 Hz, in order to prevent significant undersampling of the data which e.g. means that we can measure 100 Hz oscillations only over an area of ~20–25% of the full FOV, which correspondingly reduces the number of trackable beads (see also below).

As the modulation frequency and thus stiffness of a sample increases, the displacement decreases at a constant forces. Since a complarable displacement for each modulation frequency was desired, the force was increased with modulation frequency. However, we always ensured that applied stresses were not large enough to give rise to non-linear effects such as visible strain-stiffening.

### Choice of beads, objective and their influence on multiplexing

Whereas for active microrheology in liquids, the probe size of the bead is highly relevant, this is not the case for our samples which behave as viscoelastic solids. The sole exception is that, when probing gels, the beads have to be larger than the mesh size, such that slipping is prevented. Therefore, the reason for using rather large bead sizes (>5 μm diameter) for the cell measurements was to distribute stresses over a larger area, which might be more comparable to the in vivo conditions of the cell types studied here.

An important parameter regarding the number of trackable beads and thus multiplexing capability is the choice of the objective which defines the magnification and thus the absolute size of the FOV. Here, we used objectives with a magnification of either ×10, ×20, or ×40, corresponding to image areas of $1200 \times 1500$, $600 \times 750$, or $300 \times 375$ μm², respectively. There is an obvious trade-off between multiplexing and tracking resolution: while a smaller magnification objective allows imaging much more beads in a single FOV, the bead image will be captured by fewer pixels, which diminishes the resolution. This is the likely reason why our gel data (recorded using a 10x objective) features a larger tracking error (~50 nm) compared to e.g. RBCs (~20 nm) which were studied using a 40x objective. Nevertheless, the localization precision likely also depends on other factors, such as the bead diffraction pattern and thus the relative bead height (in case of imaging 3D samples such as gels), therefore a more detailed investigation is beyond the scope of this work.

Apart from multiplexing, we faced an additional limitation in the choice of beads when tracking beads attached to cells. The reason is that cells also scatter light and the corresponding diffraction pattern negatively interfers with the bead tracking. Therefore, we were forced to use larger magnifications for imaging cells, which resulted in reduced multiplexing. RBCs are the most concerning in this respect, since they strongly absorb light, and have a size similar to the beads, meaning the two diffraction patterns are more likely to interfere. Thus, for RBCs we used a 40x objective, and accordingly, the RBC experiments accounted for the lowest multiplexing yields (down to ~5 RBCs/FOV).

Probing gels allows for the largest amount of multiplexing, not only since this allows using the largest FOV, but also since there we track beads spread over a 3D region. Nevertheless, in this case, a substantial fraction of beads (up to 50%) was still rejected for analysis in cases when beads were

located either too close to either the surface, to the node or too far from the focus. The reason for the rejection was that in the first two cases, this would result in too low amplitudes (and consequently an overestimation of the stiffness), and in the third case in erroneous tracking. We identified these beads by first elucidating the relative height of each bead by determining the distance of their focal plane with respect to the surface, and then ensured that only beads located at a surface distance of ~5–15 μm were considered for the analysis. We finally also rejected the tracking of bead pairs which were too close to each other (<10 μm distance); furthermore, we provide control data which demonstrates that this chosen bead distance is large enough to exclude bead-to-bead interference within the data (Supplementary Fig. 1c). Nevertheless, despite this reduction the large size of the FOV still meant that up to ~100 beads could be tracked and analyzed simultaneously.

### Force-voltage calibration and field heterogeneity

In this work, we use a specific resonance frequency of the flow cell that creates a vertical acoustic standing wave with a node at a distance of about ~18 μm from the bottom of the flow cell. Compressibile particles will be drawn to this node with a force $F_B$ that depends on the squared voltage $U^2$ particle volume $V_B$ and the so-called acoustic contrast factor $\varnothing$

$$F_B = a * r^3 * V^2 * \varnothing = \beta V^2 \qquad (1)$$

which in turn depends on the particle density and compressibility ($\varnothing$) in relation to the same parameters of the solvent:

$$\varnothing = \frac{\rho_p + 2/3(\rho_p - \rho_m)}{2\rho_p + \rho_m} - \frac{1}{3}\frac{\rho_m c_m^2}{\rho_p c_p^2} \qquad (2)$$

where $\rho_p$ and $\rho_m$ being the densities, and $c_p$ and $c_m$ of particle and solvent, respectively.

While, in theory, cells will also respond to the acoustic field, their contrast factor is much lower than that of the silica beads we used for cell experiments. Additionally, the cells we used are flat and adhered to the surface, which results in the cells being less sensitive to the acoustic force towards the node. On the other hand, acoustic forces can only act on structures isolated from the surrounding solution, thus gels will not respond much to the acoustic force field, either. Overall, we can rule out considerable influence of the acoustic field on the probed structures other than the stresses exerted on the attached beads.

From the above, it is furthermore clear that applied forces are unidirectional, thus stresses can solely be applied in one direction. This means that oscillations can only be applied to a probed structure if the latter exerts a notable restoring force in opposite direction to the force field. For this reason, AFMR cannot be studied on liquids and is instead limited to solid systems. For the same reason, we cannot oscillate a sample without applying a prestress which at minimum constitutes 100% of the force oscillation amplitude.

In order to be able to assign a certain piezo input voltage to the corresponding force within a given FOV, we used the shooting beads method, as described in[36]. This procedure takes advantage of the fact that the drag force of a spherical particle in a fluid is accurately described by Stokes' law which itself contains with the drag velocity v only one parameter which is not known a priori, but can be easily measured. We then apply a range of increasing voltages (0.5–2 V) to the beads in question and are able to track their full trajectory towards the acoustic node in z-direction. By fitting the shooting-up traces, the average bead velocity can be determined. Note that we account and correct for the increased viscosity in close proximity of the glass surface using the Brenner approximation[90]:

$$\gamma_\perp = \frac{\gamma_0}{1 - \frac{9R}{8h} + \frac{R^3}{2h^3} - \frac{57R^4}{100h^4} + \frac{R^5}{5h^5} + \frac{7R^{11}}{200h^{11}} - \frac{R^{12}}{25h^{12}}} \qquad (3)$$

In the end, we average the determined force over all measured beads and voltages and plot it against the voltage (Supplementary Fig. 1a). The resulting curves then can be fitted with a parabolic function in order to yield the voltage-normalized force which is used to determine force amplitudes in the AFMR measurements. At the same time, the accuracy of the fits not only demonstrates that Newtonian fluid theory holds, but also that the used equations accurately describe the underlying physics.

It is well known that acoustic fields in standard chambers are not fully homogeneous, but that the local force can vary depending on the FOV region. This variation will mean that some beads might experience higher, other lower forces than the average value, and will accordingly result in an artificially increased spread of the data. While it is possible to reduce this artificial spread using a so-called heat-map correction procedure, we did not apply this correction to our data for two reasons. (1) From the data shown in Fig. 2, we deduced that the additional average variance introduced by the instrument is typically less than 30%, thus considerably smaller than the cell-to-cell variance we observed. We achieved this rather homogeneous force field by measuring only on central FOVs along the length of the chip by scanning the cell distribution along the chip with a 10x objective. (2) In this work, we focused on determining the variance of parameters which are not affected by this heterogeneity, thus either the exponent of the frequency-dependence, or the variance of a single cell over time (see below). Additionally, the accuracy in force determination we observed here, is similar to what has been previously observed in MT, i.e. force errors of ~20% (low f)[91]. Similar error values have also been reported for AFM height/topography determination (20–30%[92]).

### Data analysis workflow

Data analysis was done via a custom-written Python script (github link). Intensity and z-distance traces are exported from the measurement computer via our LabView software[36] and then loaded (one frequency at a time) into the Python script performing the Frequency Response Function analysis; the analysis workflow performed by the script is displayed in Supplementary Fig. 4a.

**Absolute amplitude determination.** Theoretically, our data can be described in terms of a sinusoidal input force

$$F(t) = A_F sin(\omega t) \qquad (4)$$

where $\omega$ the oscillation frequency, $t$ time and $A$ amplitude, being fed to a dynamic system, resulting in a sinusoidal output distance of the same frequency, but having a phase delay $\delta$

$$d(t) = A_d sin(\omega t + \delta) + N \qquad (5)$$

where N is a noise element.

While $d(t)$ is simply obtained from the z-tracking of the beads in the set-up, the synchronization LED gives us access to $F(t)$ in a simple manner. Thus, we correlate the average sLED intensity amplitude deduced from the chosen background ROI with the set piezo voltage, which in turn can be related back to the force using the force-voltage calibration data. Note that, while the brightness readout of the sLED intensity introduces a slight experimental noise to the force read-out (0.1%), this does not noticeably contribute to the measurement accuracy. Finally, the corresponding amplitudes of distance and force are then obtained directly from the fast Fourier transform of $d(t)$ and $F(t)$, respectively.

**Phase shift determination.** In order to extract the phase shift between force and distance data, we made use of a custom-made Python script (github.com/PLSysGitHub/AFMRv23) based on frequency response function analysis[25]. The output d(f) in Fourier space, considering the

diagram in Supplementary Fig. 4a, can be written as:

$$d(f) = H(f)F(f) + N \tag{6}$$

In the frequency domain, a linear time-invariant frequency response function $H(f)$ can then be expressed, for a system in steady-state (i.e. for which the output signal differs from the input only in amplitude and phase) via the H estimator for the case in which the noise element $N$ is assumed to be affecting only the output $y(f)$:

$$H(f) = \frac{P_{Fd}}{P_{FF}} \tag{7}$$

The above is obtained, starting from Eq. (1), by computing the cross spectral density $P_{Fd}$ between the two signals:

$$
\begin{aligned}
P_{Fd} = E\{F(f)^* d(f)\} &= E\{F(f)^* [H(f)F(f) + N]\} = H(f)E\{F(f)^* F(f)\} \\
&+ E\{F(f)^* N\} P_{Fd} = H(f)E\{F(f)^* F(f)\} P_{Fd} \\
&= H(f)P_{FF}H(f) = \frac{P_{Fd}}{P_{FF}}
\end{aligned}
\tag{8}
$$

where $E$ denotes an estimator (e.g. ensemble average), and * refers to the complex conjugate of the function. The frequency response function is therefore computed via Eq. (3) using Python's scipy.signal.csd package, by taking the ratio between $P_{Fd}$ - the cross spectral density of $F(f)$ and $d(f)$ - divided by $P_{FF}$, the auto spectral density of the input $F(f)$. The resulting frequency response function H(f) is a complex number, made of a real H' and imaginary part H", whose phase directly reports on the phase shift $\delta$ between F(t) and d(t). Thus, by using the complex-plane relation between H' and H":

$$\delta = tan^{-1} \frac{H''}{H'} \tag{9}$$

**Calculation of the viscoelastic moduli.** The procedures above determine absolute force and distance amplitudes $A_F$ and $A_d$, as well as the phase shift $\delta$ in the complex plane; these values can be translated into storage and loss moduli using either of the two procedures below:

1. For a bead fully embedded in a collagen gel, G' and G" are given as:

$$G' = \frac{A_F}{6\pi RA_d} cos\delta \tag{10}$$

$$G'' = \frac{A_F}{6\pi RA_d} sin\delta \tag{11}$$

where R is the radius of the probing microspheres[22]

2. For a bead on the cell surface (RBC or HgF), a very similar formula is used, however employing an additional geometric correction term b(θ) which depends on the bead contact angle θ:

$$G' = \frac{A_F}{b(\theta) * 6\pi RA_d} cos\delta \tag{12}$$

$$G'' = \frac{A_F}{b(\theta) * 6\pi RA_d} sin\delta \tag{13}$$

The correction term is defined by $b(\theta) = \left(\frac{9}{4\sin\theta} + \frac{3\cos\theta}{2\sin^3\theta}\right)^{-1}$, and in our case we assume $\theta \approx 30°$[31,42].

The raw recorded traces were observed individually and filtered out if they:

a. Showed error in frequency response function analysis, i.e. incorrect frequency peak due to noisy trace (position of the peak not matching input frequency);
b. Showed errors in tracking (due to interferences in LUTs);
c. Showed too small/noisy amplitude (<20 nm, which is the z-resolution of the instrument on these experiments[40]);
d. Showed strong non-linearities in the Lissajous plots;
e. Showed ruptures or other inconsistencies in the traces, such as slipping. These features are directly identifiable from the raw traces, as in both cases the ruptured/slipped bead would then move towards the node, resulting in a linear shooting-up trajectory, z > 10 µm.

### HgF structural damping fit
The fibroblast data was fitted with a structural damping model, as:

$$G' = G_0 \left(\frac{\omega}{\omega_0}\right)^\alpha \tag{14}$$

$$G'' = G_0 \tan\left(\frac{\pi\alpha}{2}\right) * \left(\frac{\omega}{\omega_0}\right)^\alpha + \omega\mu \tag{15}$$

where $G_0$ is the low-frequency plateau modulus at $\omega_0$ ($\omega_0$ is taken as 1 Hz[32]), $\alpha$ is the power-law exponent, and $\mu$ a Newtonian coefficient.

### Collagen (Type I) network experiments
First, the AFS flow cell was passivated with 0.1% casein (9005-46-3; Sigma Aldrich). Then, we assembled collagen gels in a high salt buffer of 60 mM phosphate buffer, pH 7.4 (10028-24-7; Sigma Aldrich and 7558-86-7; Fluka) and 300 mM NaCl (S9888; Sigma Aldrich), similar to previous experiments[49]. For the low concentration gel assembly, we subsequently mixed 40 µl of buffer, 2.5 µl of a 6.39 µm diameter polystyrene beads suspension (1% w/v; PS-COOH-B895; Micro particles GmbH, suspended in a solution containing 0.05% casein for passivation purposes), 0.3 µl NaOH (0.4 M; 1310-73-2; Riedel-de Haën, in order to neutralize the acetic acid content of the collagen stock) and 3.5 µl of 3 mg/ml type I rat tail collagen stock (A1048301; Thermofisher) solution on ice. Note that the main reason behind the large bead diameter was to prevent them slipping through the gel pores which indeed we never observed. The sample was then vortexed for ~10 s and inserted into the AFS chamber, before the inlet/outlet channels were blocked with concentrated sucrose solution to prevent evaporation. Then, the sample was first placed ~100 s upside down in a 37 °C oven, and subsequently turned around and incubated a further 15 min at 37 °C. This particular incubation protocol was chosen, since in a standard incubation ( = upright incubation from the start) >90% of the beads ended up sedimented to the surface.

For the high concentration assembly, we chose the same conditions with the difference that we added 0.9 µl of NaOH and 10.5 µL of collagen stock. During the assembly of the collagen beads sedimented due to gravity, thus the majority of beads was on the glass surface; however, we only measured fully embedded beads slightly higher (~5–15 µm) in the flow cell and recorded a LUT at the focal planes to that height. The resonance frequency was adjusted after the collagen was assembled in the chip, as the change in density from water shifts it slightly higher, by approximately 0.05 MHz.

### RBC experiments
RBC data ($N = 143$) was collected over $N = 10$ days of experiments, and blood was collected from $N = 4$ donors (28 F, 25 F, 27 M, 30 F). Briefly, approximately 20 µL of blood was collected from healthy volunteers by finger pricking (iHealth ALD-602 glucose meter), spun down at 200 g for 2 min, and finally diluted in 200 µL Ringer's buffer, consisting of 125 mM NaCl, 5 mM KCl (60130; Fluka), 1 mM CaCl$_2$ (10035-04-8; Sigma Aldrich), 1 mM MgSO$_4$ (63072; Fluka), 5 mM d-glucose (49161; Fluka) and 32 mM HEPES (7365-45-9; Sigma Aldrich). Microspheres were functionalized with Concanavalin A (11028-71-0; Sigma Aldrich), as previously described[40].

ConcanavalinA is routinely used as a non-specific bead coating[34,93] and binds to the glycoproteins in the plasma membrane. In short, 6.59 μm silica microspheres (SiO$_2$-R-SC33-1; Micro particles GmbH) were washed in milliQ at $500 \times g$ x 2 min, resuspended in 1 mL of 3% HCl (7647-01-01; Sigma-Aldrich) for surface activation (10 min), washed again and incubated in Concanavalin A for 30 min at 4 °C. Finally, the bead pellet is resuspended in 500 μL of Ringer's buffer before flushing them in the AFS chip.

Prior to the measurements, the AFS flow cell was treated with 0.1% poly-L-lysine (25988-63-0, Sigma Aldrich) for 30 mins to promote cell adhesion. After flushing the RBCs in the chamber, they are left to adhere for ~2 mins; then, beads are flushed in and sediment on the RBCs. All measurements were performed at 37 °C by making use of an AFS temperature control module (LUMICKS BV), see Supplementary Fig. 4b. For RBCs, approximately 20 cells can be measured simultaneously at full FOV, while the number decreases five-fold when measuring at high frequencies (f$_{mod}$~100 Hz). Note also that the number of trackable beads is dependent on the success rate of bead-cell attachment. We perform all measurements at modulation depth of 50% in order to avoid non-linear effects.

### Human Gingival Fibroblasts experiments

The HgF cell line was a kind gift from T. De Vries (ACTA, Amsterdam), and extracted from the residual gingiva on wisdom tooth of a healthy 20 years old female. Cells were grown in T-75 flasks at 37 °C in a 5% CO$_2$ environment. Cells were cultured in DMEM medium, supplemented with high glucose, sodium pyruvate (31966-021; Gibco, Life Technologies) 10% FBS (10270-106; Gibco, Life Technologies), 25 mM HEPES (pH 7.2; 15630-080; Gibco, Life Technologies), 1% penicillin/streptomycin (15140-122; Gibco, Life Technologies) and 1% non-essential amino acids (11140-050; Gibco, Life Technologies). Culture medium was exchanged every 3 days, and cells were passaged after reaching ~80/90% confluence.

The day before the experiments, the AFS flow cell was incubated overnight with 70 μg/ml rat tail collagen I (A1048301; Gibco, Life Technologies) diluted in 200 mM acetic acid (33209; Sigma Aldrich). On the day of the experiment, cells were harvested from the flasks after 4 min incubation in 2 mL TrypLE™ Express (12604-013; Gibco, Life Technologies), harvested by centrifugation ($200 \times g$ x 5 min) and resuspended in 1 mL of culture medium. Cells were then flushed in the AFS flow cell and incubated at 37 °C + 5% CO$_2$ for at least 2.5–3 h in order to allow for surface adhesion. At the start of the experiments, new medium was flushed in the chamber. Then, concanavalin A-coated 6.59 μm silica beads (see above) were flushed in, which sedimented randomly over the cells. Large silica beads were used as they allowed us to apply >400pN force on fibroblasts, which are a rather stiff sample. HgF data ($N = 71$) was collected over $N = 3$ days of experiments for cells at passages between 6 and 8. To decrease data-collection time to below 10 min while retaining a relatively large imaging area (~25% FOV), we limited the measured frequency range to 0.1–30 Hz. For the cytochalasin D experiments, 100 uL of culture medium supplemented with 500 nM cytochalasin D was flushed into the chamber and cells were left to incubate for about 30 min before measurements.

### Statistics and reproducibility

The 0.2 mg/mL collagen gel (Fig. 2c, d, 3a, c, d, Supplementary Fig. 1f) has been measured two times, resulting in a total of 181 beads used. The 0.6 mg/mL collagen gel (Fig. 3b–d, Supplementary Fig. 2c) has been measured two times, resulting in 139 beads used.

The rheology measurements for RBCs (Fig. 4a) have been executed 11 times, resulting in 143 beads. The time-series (Fig. 4b) is executed one time, and has 5 beads at 0 min, 5 beads for 20 min, 6 beads for 40 min and 6 beads for 60 min. The modulation depth measurement (Fig. 4c, Supplementary Fig. 2d) is executed two times and has 24 beads used for 0.1 Hz, three times resulting in 19 beads for 1 Hz and two times resulting in 17 beads for 10 Hz. The pre-stress amplitude measurement (Fig. 4d, Supplementary Fig. 2e) is executed four times resulting in 14 beads for 0.1 Hz, six times resulting in 19 beads for 1 Hz and four times resulting in 10 beads for 10 Hz.

HgF cells without cytochalasinD (Fig. 4e) were measured three times, resulting in 71 beads. With cytochalasinD (Fig. 4f) cells were measured two times (same as without), resulting in 32 beads.

The two-tailed paired $t$ test in Fig. 4f has four degrees of freedom, a t of 3.856 and an exact $p$ value of 0.0182.

### Separating instrument noise from intrinsic sample variance

One of the aims of this work is to use AFMR to elucidate the heterogeneity of a probed sample by quantifying relevant biophyscial parameters. This, however, requires that we are able to separate the intrinsic sample variance from external factors, most notably instrument-related noise regarding the measured parameters. Nevertheless, the latter is not straightforward to determine; thus, whereas we can easily quantify direct noise sources such as the bead tracking precision, there is no simple way to predict how this will influence the final measurement parameter of storage and loss moduli.

For this reason, we use an empirical way to estimate an upper bound for this influence and do this by repeatedly measuring the same beads embedded in a collagen gel. We then assume that the resulting variance yields an upper bound for the measurement-related error. The latter is quite important to consider, since we have no independent means to verify that the sample mechanics will not change over time. This approach has the particular advantage that - by re-measuring on the same beads - many noise sources such as field heterogeneity, bead polydispersity and other position-related noise will not contribute to the deduced variance.

We determine instrument-related variances for two different measurement parameters: in one case, we compare relative s.d. values for the moduli (as shown in Fig. 2c, d), in the second case, we determine the variance of the G"-exponent which describes the frequency dependence of the moduli (as shown in Supplementary Fig. 1f). The latter is of particular interest when comparing the bead-to-bead variance within a sample. The reason for this is that according to Eq. 9–12, calculated moduli depend linearly on the force, thus will be directly affected by the above mentioned error sources, such as field heterogeneity and bead polydispersity, since those affect the applied forces. However, there is no reason to assume that a slight variance of the force will affect the frequency dependence of the moduli, in particular since we are probing the linear regime. This is the reason why we deduce from the fact that the bead-to-bead variance of the G" collagen exponents is much larger than upper bound of the corresponding instrument noise that the distribution reflects mainly the true intrinsic heterogeneity of the sample. Note that we in this case focus on the G"-exponent for the simple fact that this is more than two times larger than the G'-exponent (Fig. 3a), whereas the estimated G"-variance is less than twice that of G'.

We come back to the instrument error discussion when considering the distribution of G"-exponents of the RBC data derived from single beads (Fig. 4a and Supplementary Fig. 2c). In this particular case,—similar to the corresponding collagen data discussed in Fig. 3d—we compare the RBC data variance of the G"-exponent to the data in Supplementary Fig. 1f. From the relatively larger data spread of the RBC data, we then likewise conclude that this reflects an intrinsic samples heterogeneity. At first sight, it might seem surprising that we are trying to relate the results from two different samples (RBC and collagen), since the choice of a sample might also affect the instrument resolution. However, we posit that we can still use the collagen reproducibility as an upper bound for the RBC data for the following reason: if we compare raw exemplary traces of collagen and RBC data (Fig. 2a and Supplementary Fig. 2a), we note in the first case a significantly larger tracking noise, in particular at higher, but also lower frequencies. Since it seems reasonable to assume that the tracking noise plays a role in determining the noise of our modulus (and therefore the corresponding exponent), we think it safe to assume that the RBC G"-exponent reproducibility should not be much larger than the 6% we estimated in the case of collagen. While in theory, one could also try to deduce the corresponding RBC value directly from repeated measurement of RBCs, we think this will not yield a proper estimate of the instrument noise since RBCs are known to display intrinsic dynamic variance of the mechanic response on longer timescales.

It has been well established that cell stiffnesses measured with MT and AFM feature large variance[68]. The spread in the data we report here obtained using AFMR is in line with these earlier studies. For example, relative errors of ~35%[62], ~46%[34] and ~41%[94] were reported for OT, MT and AFM experiments on RBCs. Similar spreads in data were also obtained for adherent, fibroblast cells (~50%[68,95]) as well as adherent muscle cells[85].

## Reporting summary

Further information on research design is available in the Nature Portfolio Reporting Summary linked to this article.

## Code availability

The data acquisition code is available via Zenodo with https://doi.org/10.5281/zenodo.11115841[96], and the data analysis code is available on Github: github.com/PLSysGitHub/AFMRv23.

## Data availability

The source data named 'Acoustic Force Microrheology (AFMR)' can be downloaded from https://dataverse.nl/dataset.xhtml?persistentId=doi:10.34894/SEJA5T[97] and the figure source data is available in the Supplementary Data 1.

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

## Acknowledgements

We thank T. de Vries (ACTA) for providing the HgF cells. Furthermore, we thank Tristan den Blanken from Elektronica Bèta (VU) for helpful discussions regarding instrument synchronisation. H.W. thanks the Deutsche Forschungsgemeinschaft for financial support (WI 5434/1-1). Moreover, G.W. and G.B. acknowledge support by the Netherlands Organisation for Scientific Research (NWO/OCW), as part of the BaSyC Gravitation program. K.H.T acknowledges support by Netherlands Organisation for Scientific Research (NWO/OCW), file number: 680-91-102.

## Author contributions

GB and KHT contributed equally. GB conceived the study; Methodology, GB and KHT designed the methods; GB, KHT, XMRS and HW wrote (analysis) software; GB, KHT, ASB and HW performed validation of the method, data analysis and wrote the original draft; GB, KHT, ASB and XMR acquired the data; EJGP, GJLW supervised the project and acquired the funding; All authors reviewed and edited for the final version.

## Competing interests

The authors declare the following competing interest: AFS technology intellectual property is owned by LUMICKS b.v. EJGP and GJLW declare financial interest in LUMICKS.
