## [Peer Review File · Communications Biology]

Reviewers' comments:

Reviewer #1 (Remarks to the Author):

Review

In this research article, the team has extended the use case for the AFS by smartly incorporating an extra LED into the illumination pipeline. They showcase the system by measuring the viscoelastic properties of collagen, and a variety of cells. I really enjoyed reading this paper. There is a lot here. I would recommend a follow-up methods paper to increase uptake of this technique with those groups that are actively using the AFS.

General Comments:

Line 125 to 127: Nevertheless, we do not have indications that the resulting larger variance of absolute moduli significantly affects average.

Can you provide numbers to support this claim? Is the magnitude of the "large variance" of the absolute moduli acceptable and how does it compare to MT and AFM?

Line 242: This is confirmed by the observation that the corresponding relative s.d. values of RE(ahigh) \approx 50% and RE(alow) \approx 18% are much higher than the upper bound of the instrument error (6%)
The rationale for deducing the relative instrument error makes logical sense. That being said, does the above imply that the relative error in measuring the exponents is as high as 50%? If yes, what does that mean for being able to tease apart statistically significant differences between samples? If yes, what can be done to reduce this error?

Figure 4:

4C and D: Please plot the full extent of the error bars (cut off)

4F: Can you plot the errors bars. Can you also comment on whether the difference observed is statistically significant. A 2-fold decrease is meaningful, what are the 95% confidence intervals around this? Is the error for a reported here in SD or SEM?

Also, including all the sample sizes in the statistics section is nice, however I found myself going back and forth between the figures and the statement to look up sample size. For convenience, please include n whenever giving measurements within the figures, helps with readability.

How long were the cells incubated with Cytochalasin D?

Line 335: multiplexing/parallel measurements comparable to MT, while it can apply high forces (up to tens of nN) similar to AFM.

How does this new technique compare in terms of accuracy or repeatability?

Line 341: While microrheology data on single beads have been reported before, our results are noteworthy since they can assign a confidence interval to the data, which is vital to identify a significant deviation of viscoelastic parameters between different datasets.

What does this mean? Where have confidence intervals been discussed?

Line 392: on location and exposure time, we choose for every new FOV a new ROI and recalculate the corresponding brightness-to-force conversion factor.

How did you return to these FOV's or were they established on the fly? So, if one wanted to measure the elasticity of HgF cells, the FOV can be calculated on the fly or has been pre-registered from before?

Line 449: by first elucidating the relative height of each bead by determining the distance of their focal plane with respect to the surface..

How was the surface location determined or z=0 determined?

Line 486: Note that we account and correct for the increased viscosity in close proximity of the glass surface using the Brenner approximation.
Please provide the equation and reference.

Line 496: While it is possible to reduce this artificial spread using a so called heat-map correction procedure, we did not apply this correction to our data for two reasons. 1) From the data shown in Fig. 2, we deduced that the additional average variance introduced by the instrument is typically less than 30%, thus considerably smaller than the cell-to-cell variance we observed. We achieved this rather homogeneous force field by measuring only on central FOVs along the length of the chip. 2) In this work, we focused on determining the variance of parameters which are not affected by this heterogeneity, thus either the exponent of the frequency-dependence, or the variance of a single cell over time.

Im trying to understand the logic in this statement. For 1), it is stated that the instrument introduces an error of less than 30%, can you also include what was the observed cell-to-cell variance %? An inherent error of 30% is large, would it not be mitigated by reducing FOV size, better FOV localization, characterization etc? Or is this error sufficient to still deduce statistically significant differences between samples?

Line 500: We achieved this rather homogeneous force field by measuring only on central FOVs along the length of the chip.

How did you ensure that you were in the centre the whole time?

Line 563:

From reference 22, G' is the storage modulus, not the elastic modulus. Its correctly referenced in Line 264 as the storage modulus, but is labelled as the elastic modulus within the figure. As far as I am aware, the two are not interchangeable and measure different phenomena. Unless the frequencies used in this study are high enough to justify the equality?

Reviewer #3 (Remarks to the Author):

The authors apply acoustic force spectroscopy to measurements of active microrheology in cells, and more specifically of red blood cells and fibroblasts. The main advantage of the method is the capability to parallelize measurements over cell populations to characterize the heterogeneity of their viscoelastic response. This is a useful contribution to the community investigating cell mechanics. The paper is well-written but needs some clarifications. I am happy to recommend the paper for publication if the following comments are addressed.

1. I found that the method, as presented, does not show any breakthrough or discovery when applied to red blood cells or fibroblasts. The tone of the paper remains mostly methodological, sometimes emphasizing the heterogeneity of the samples but without identifying a new result. Could the authors underline a discovery obtained from AFMR compared to previous measurements with other techniques?

2. AFMR is presented as a tool for active measurements where an oscillatory driving force is used to measure the elastic G' and viscous G'' moduli of cells. Can acoustic force spectroscopy be used to measure time correlation functions or power spectra? This is important to discriminate active from passive fluctuations in cells, an important and currently hot topic in the field.

3. The results for the collagen I gels (figure 3) show that the power law exponents for G'' and G' ,

$\alpha G''$ and $\alpha G'$ do change between high and low concentrations (e.g. panel 3D) but their ratio $\alpha G''/\alpha G'$ is approximately 2.5, i.e., the same for low and high concentrations. Is there an explanation for that?

4. Related to the previous question, figure 3C (inset) shows the variability of $\alpha G''$ from bead to bead. Does $\alpha G'$ also vary from bead to bead? Does the ratio $\alpha G''/\alpha G'$ remain approximately constant and equal to the above value of 2.5?

5. The results for G' and G'' in red blood cells in Figure 5A are fitted to power laws, but the results for the viscous modulus clearly show two slopes above and below 1Hz. Also for G' two values for $\alpha G'$ are hinted at low and high frequencies. This could be taken as evidence of two separated relaxational processes below and above 1Hz. In fact, for red blood cells, at least two relaxational steps have been identified between 0.01-0.1 Hz and above 1Hz. Ultimately, the assumption of a single-exponent power-law dependence for G' and G'' might not be true. The authors should comment on this.

6. Figure 4C for 10Hz shows too large error bars that are unphysical. To avoid this, error bars should be estimated over $\log|G^*|$ rather than $|G^*|$

7. On page 11, G' values of RBCs at 0.01Hz are equal to 72(4) Pa (Figure 4A) at the lower end of other studies reporting values in the kPa range (Refs 67,69,70). How do you explain this discrepancy? How do these G' values relate to the stiffness values of 5-10pN/um reported in optical tweezer studies (e.g. Refs 71,74)?

8. In the discussion it is said (line 360) that AFMR is suitable to investigate the non-linear mechanical properties. Why non-linear effects are important? Any intuition about their biological role?

9. In Figures 1B and 1C, color codes purple and green for force and distance appear exchanged.

10. Some acronyms should be defined, for example, AFMR or FOV (field of view). AFMR is defined in the title but should be in the text also the first time it appears in the text.

Reviewers' comments:

Reviewer #1 (Remarks to the Author):

Review

In this research article, the team has extended the use case for the AFS by smartly incorporating an extra LED into the illumination pipeline. They showcase the system by measuring the viscoelastic properties of collagen, and a variety of cells. I really enjoyed reading this paper. There is a lot here. I would recommend a follow-up methods paper to increase uptake of this technique with those groups that are actively using the AFS.

General Comments:

Line 125 to 127: Nevertheless, we do not have indications that the resulting larger variance of absolute moduli significantly affects average.

Can you provide numbers to support this claim? Is the magnitude of the "large variance" of the absolute moduli acceptable and how does it compare to MT and AFM?

We thank the reviewer for pointing out that the statement in the manuscript was lacking appropriate referencing. We now added in the main text (lines 124-128) a reference to previous work (Bogatyr et al., 2022), where the heterogeneity of the acoustic field was found to not significantly affect mechanical measurements for cell samples similar to the ones studied here. Furthermore, we also refer the reader to lines 520-522 of the Methods section for comparison of force-determination accuracy between AFS and MT/AFM. We added a comparison of the variance in moduli obtained with similar methods to lines 732-736 of the Methods, in the section 'Separating instrument noise from intrinsic sample variance'.

Line 242: This is confirmed by the observation that the corresponding relative s.d. values of $RE(\text{high}) \approx 50\%$ and $RE(\text{low}) \approx 18\%$ are much higher than the upper bound of the instrument error (6%)

The rationale for deducing the relative instrument error makes logical sense. That being said, does the above imply that the relative error in measuring the exponents is as high as 50%? If yes, what does that mean for being able to tease apart statistically significant differences between samples? If yes, what can be done to reduce this error?

In the case of figure 3D, the large relative errors on the exponents stem from the heterogeneity of the sample itself rather than an instrument error. By measuring the same beads over time (Figure 2C) we demonstrated that the instrument error can account for 6%, which in fact represents the error in determining an individual exponent (wording now clarified in the main text).

In addition, we agree with the reviewer that a statistical significance test was missing to demonstrate that, although highly heterogeneous, the samples can be discriminated. We now added the statistical test to the caption of Figure 3D ($p < 0.0001$).

Figure 4:

4C and D: Please plot the full extent of the error bars (cut off)

We now changed the format of the corresponding figures as suggested.

4F: Can you plot the errors bars. Can you also comment on whether the difference observed is statistically significant. A 2-fold decrease is meaningful, what are the 95% confidence intervals around this? Is the error for α reported here in SD or SEM?

As we used the SEM, these values were so small at frequencies lower than 30 Hz, that the error bars would not be visible when overlapped with the marker. We now changed the format of the marker which should allow SEM error bars to be more clearly visible. We also did the same for Figure 4A and Figure 3A/B. We also performed a statistical test in order to add confidence intervals to the caption and a paired t-test of the G^ with and without cytochalasin D where a p-value of 0.0182 was obtained.*

Also, including all the sample sizes in the statistics section is nice, however I found myself going back and forth between the figures and the statement to look up sample size. For convenience, please include n whenever giving measurements within the figures, helps with readability. How long were the cells incubated with Cytochalasin D?

As suggested, we now added the statistics to the figure captions to help the reader.

Regarding the cytochalasin D incubation, we added more information regarding incubation times and concentration to the corresponding Methods section, lines 668-670.

Line 335: multiplexing/parallel measurements comparable to MT, while it can apply high forces (up to tens of nN) similar to AFM. How does this new technique compare in terms of accuracy or repeatability?

As stated in the reply to the first point, we have now added two small sections to the Methods respectively providing: 1) a comparison of accuracy in determination of force with MT and AFM, and 2) a comparison of the variance in the complex moduli obtained using these techniques. Unfortunately, there are no studies where the reproducibility of the complex modulus is thoroughly discussed as we have done by means of repeated measurements in Figure 2.

Line 341: While microrheology data on single beads have been reported before, our results are noteworthy since they can assign a confidence interval to the data, which is vital to identify a significant deviation of viscoelastic parameters between different datasets.

What does this mean? Where have confidence intervals been discussed?

We thank the reviewer for the comment and agree that the current phrasing might be unclear. The confidence interval that we refer to is simply the quantification of the reproducibility (Figure 2D, RE < 20%), which allows us to define an upper bound for the instrument error. We now changed the wording in the main text and included a reference to a corresponding note on error quantifications in the Methods.

Line 392: on location and exposure time, we choose for every new FOV a new ROI and recalculate the corresponding brightness-to-force conversion factor.

How did you return to these FOV's or were they established on the fly? So, if one wanted to measure the elasticity of HgF cells, the FOV can be calculated on the fly or has been pre-registered from before?

We agree with the reviewer that the phrasing of our method might be unclear. In order to read out the oscillations of sLED intensity, we do the following: 1) we select a measurement FOV containing the cells of interest, and 2) we select a small ROI in the same FOV which will be used solely for sLED brightness readout. Thus, there is no need of switching between sLED-ROI and FOV as the ROI can be selected within the same FOV used for mechanical probing. We clarified our description of this procedure in the corresponding Methods section (line 406-407).

Line 449: by first elucidating the relative height of each bead by determining the distance of their focal plane with respect to the surface. How was the surface location determined or $z=0$ determined?

The location of the surface was found as follows: we find the edge of the flow cell before each measurement and place it into focus, setting this as $z=0$. We then proceed to create look-up tables (LUTs) for the beads in the FOV. These LUTs served as a template to match each bead in the measurements to their corresponding z height, so then we are able to select only beads at specific heights (in our case 5-15 μ m).

Line 486: Note that we account and correct for the increased viscosity in close proximity of the glass surface using the Brenner approximation.

Please provide the equation and reference.

We now added the Brenner equation and reference to the corresponding Methods section (lines 502-503).

Line 496: While it is possible to reduce this artificial spread using a so called heat-map correction procedure, we did not apply this correction to our data for two reasons. 1) From the data shown in Fig. 2, we deduced that the additional average variance introduced by the instrument is typically less than 30%, thus considerably smaller than the cell-to-cell variance we observed. We achieved this rather homogeneous force field by measuring only on central FOVs along the length of the chip. 2) In this work, we focused on determining the variance of parameters which are not affected by this heterogeneity, thus either the exponent of the frequency-dependence, or the variance of a single cell over time.

Im trying to understand the logic in this statement. For 1), it is stated that the instrument introduces an error of less than 30%, can you also include what was the observed cell-to-cell variance %? An inherent error of 30% is large, would it not be mitigated by reducing FOV size, better FOV localization, characterization etc? Or is this error sufficient to still deduce statistically significant differences between samples?

We thank the reviewer for pointing out this unclarity. The observed cell-to-cell variance is expressed as relative error in Table1, which we did not reference correctly in the text. This has now been corrected.

Reducing FOV size does not guarantee a minimization of the spread in field heterogeneity as this is location-dependent. It would, however, substantially reduce the statistics. In terms of localization, we try to reduce the spread by measuring directly below the piezoelectric element (see answer below).

Furthermore, our results based on average-fitting of cell data still allowed discrimination between wild-type and CytoD-treated fibroblasts, as well as hard and soft collagen. We now added statistical tests on the complex moduli for both collagen and cells. A paired t-test on G^ gave a p-value of 0.0182, confirming that the differences we observed between samples are indeed significant.*

Line 500: We achieved this rather homogeneous force field by measuring only on central FOVs along the length of the chip. How did you ensure that you were in the centre the whole time?

We typically start a measurement by scanning the cells in the chip with a 10x objective. This ensures selecting FOVs that are directly below the piezoelectric element that generates the standing waves. This has now been clarified in the corresponding Methods section (line 517).

Line 563:

From reference 22, G' is the storage modulus, not the elastic modulus. Its correctly referenced in Line 264 as the storage modulus, but is labelled as the elastic modulus within the figure. As far as I am aware, the two are not interchangeable and measure different phenomena. Unless the frequencies used in this study are high enough to justify the equality?

We thank the reviewer for pointing this out. Indeed G' should be referenced as the storage modulus. We now correct this throughout the manuscript to ensure consistency.

Reviewer #3 (Remarks to the Author):

The authors apply acoustic force spectroscopy to measurements of active microrheology in cells, and more specifically of red blood cells and fibroblasts. The main advantage of the method is the capability to parallelize measurements over cell populations to characterize the heterogeneity of their viscoelastic response. This is a useful contribution to the community investigating cell mechanics. The paper is well-written but needs some

clarifications. I am happy to recommend the paper for publication if the following comments are addressed.

1. I found that the method, as presented, does not show any breakthrough or discovery when applied to red blood cells or fibroblasts. The tone of the paper remains mostly methodological, sometimes emphasizing the heterogeneity of the samples but without identifying a new result. Could the authors underline a discovery obtained from AFMR compared to previous measurements with other techniques?

We present here a key technical improvement compared to previous AFS-based microrheology studies: the implementation of a synchronization scheme between force application and distance readout based on a LED, which allows high frequencies to be probed reliably. Furthermore, in terms of methodological advancements, we intended to provide a thorough error analysis and validation of single-particle measurements, which had not been presented before. Regarding the choice of samples, we wanted to showcase how the method can be applied to single cells, polymer networks and motile primary cells, to highlight the orders of magnitudes of stiffness that the method can probe. We also demonstrate how the multiplexing capabilities of our instrument can be exploited to quantify sample heterogeneities in polymer gels. Before, this had only been studied using optical tweezers, yielding limited statistics not allowing any distribution-analysis (Figure 3D). From the biological point of view, we used our method to characterize the mechanical properties of a motile primary fibroblast cell line which was not mechanically characterized before. We tried to make this now clearer in the Discussion section (line 350-352, 354-355).

2. AFMR is presented as a tool for active measurements where an oscillatory driving force is used to measure the elastic G' and viscous G'' moduli of cells. Can acoustic force spectroscopy be used to measure time correlation functions or power spectra? This is important to discriminate active from passive fluctuations in cells, an important and currently hot topic in the field.

In the current configuration, AFMR is a technique that requires the cells to be attached to the surface with probing beads on top. Similar to MT and OT, it would be possible to follow a bead on the cell surface without force application, thus tracking the MSD of the particles. This would allow for measurements of passive fluctuations as well. In terms of resolution, AFS can provide <5 nm XY resolution when tracking 1.5 μ m silica beads (Sitters et al., 2015), thus would in principle be able to resolve such fluctuations (distances 0-200nm; Turlier et al., 2018). We added this in the discussion, lines 376-378.

3. The results for the collagen I gels (figure 3) show that the power law exponents for G'' and G' , $\alpha G''$ and $\alpha G'$ do change between high and low concentrations (e.g. panel 3D) but their ratio $\alpha G''/\alpha G'$ is approximately 2.5, i.e., the same for low and high concentrations. Is there an explanation for that?

This is certainly an interesting observation of the reviewer. We carefully examined the literature and could not find any theory that would explain the conservation of the exponent ratio in the two collagen concentrations. So, at this point we do not really have an explanation for this behavior. On the other hand, single beads do not show conservation of the exponent (see answer below).

4. Related to the previous question, figure 3C (inset) shows the variability of $\alpha G''$ from bead to bead. Does $\alpha G'$ also vary from bead to bead? Does the ratio $\alpha G''/\alpha G'$ remain approximately constant and equal to the above value of 2.5?

The inset of Figure 3C in fact already shows the variability of $\alpha G'$ from bead to bead, so indeed this varies as well. In the heterogeneity discussion that follows, we focus on G'' , the parameter we believe is of higher interest. The ratio between $\alpha G''/\alpha G'$ is, for a random selection of beads, such as the ones displayed in Figure 3C, not equal to 2.5 but

oscillates between 1 and 2. This seems to indicate that the average $\alpha_{G''}/\alpha_{G'}$ ratio is not constant between single beads.

5. The results for G' and G'' in red blood cells in Figure 5A are fitted to power laws, but the results for the viscous modulus clearly show two slopes above and below 1Hz. Also for G' two values for $\alpha_{G'}$ are hinted at low and high frequencies. This could be taken as evidence of two separated relaxational processes below and above 1Hz. In fact, for red blood cells, at least two relaxational steps have been identified between 0.01-0.1 Hz and above 1Hz. Ultimately, the assumption of a single-exponent power-law dependence for G' and G'' might not be true. The authors should comment on this.

We thank the reviewer for bringing up this important point. We have decided to fit the power law at $f > 1$ Hz to allow direct comparison with previous reports (Puig-de-Morales et al., 2007). Nevertheless, we agree that our data might hint to different relaxation timescales for RBCs. We now mention this in the main text (lines 276-278) and provide references to recent papers showing evidence of this.

6. Figure 4C for 10Hz shows too large error bars that are unphysical. To avoid this, error bars should be estimated over $\log|G^*|$ rather than $|G^*|$

We thank the reviewer for pointing this out. Estimating logarithmic errors for G^ led to very small error-bars which affected readability. In order to better clarify the spread of the data (and outliers) we now changed the corresponding figure to a more clear box plot format.*

7. On page 11, G' values of RBCs at 0.01Hz are equal to 72(4) Pa (Figure 4A) at the lower end of other studies reporting values in the kPa range (Refs 67,69,70). How do you explain this discrepancy? How do these G' values relate to the stiffness values of 5-10pN/um reported in optical tweezer studies (e.g. Refs 71,74)?

Our results should be more directly comparable to AFM or MT studies, given that they share the same experimental configuration, i.e. vertical pulling on surface-attached cells. When compared to AFM studies, our results are on the 'softer' end. This could be due to the stronger surface attachment protocol used in some of the AFM studies (4% formaldehyde), as well as different probe contact areas (Wu et al., 2018). Dependence of the stiffness on the degree of surface attachment in AFS was reported previously for different concentration of poly-L-lysine, which shows higher stiffness for higher poly-L-lysine concentration (2x increase for 2x PLL; Sorkin et al., 2018). We have added this to the interpretation - lines 269-272.

Comparing our values to the stiffnesses reported in OT experiments is not as straightforward since 1) a different pulling geometry is used: vertical stretching in AFS vs. axial in OT and 2) cells are often measured in solution, such that surface attachment has no effect.

If we consider that we need to apply ~ 150 pN to produce an elongation of ~ 1 um (see SI Figure 2), while OT studies produce ~ 2 um elongations with ~ 40 pN forces (ref 41, where

comparable bead sizes are used), this confirms that pulling vertically in our configuration generates a ~7.5x stiffer response than pulling axially.

We can compare the complex modulus from ref 74, i.e. $K^ = 20 \text{ pN/um}$ to our RBCs complex modulus $G_{AFS}^* = 100 \text{ Pa}$. Using the prefactor of equation 11 (Methods) and considering a bead size of $2.5 \mu\text{m}$ to estimate the correspondent G_{OT}^* , we obtain $G_{OT}^* = 10 \text{ Pa}$. Hence, our results are one order of magnitude stiffer than the reported values, a discrepancy which can be explained by the above observations. Additionally, it is also important to note that our exponent values are in line with the results of Ref 71,74, which also report exponents ~ 0.5 .*

8. In the discussion it is said (line 360) that AFMR is suitable to investigate the non-linear mechanical properties. Why non-linear effects are important? Any intuition about their biological role?

We mention the non-linear mechanical properties as AFMR is able to apply large pre-stresses, hence making it a useful tool for this type of experiments in e.g. polymer networks (Broedersz et al., 2013, Majumdar et al., 2018). Probing of non-linear mechanics is of particular interest in soft matter as this can identify complex mechanical behaviours such as ruptures or re-adjustment of underlying cellular components, as well as providing insight into active cellular processes.

9. In Figures 1B and 1C, color codes purple and green for force and distance appear exchanged.

We thank the reviewer for pointing this out. We have corrected this.

10. Some acronyms should be defined, for example, AFMR or FOV (field of view). AFMR is defined in the title but should be in the text also the first time it appears in the text.

We thank the reviewer for pointing this out. We now define the acronyms at lines 79 (AFMR) and 100 (FOV).

REVIEWERS' COMMENTS:

Reviewer #1 (Remarks to the Author):

My concerns/questions have been addressed.

Reviewer #3 (Remarks to the Author):

I could go in detail through the answers to my concerns provided by the authors and I am satisfied with their response. I am happy to recommend the paper for publication.